# Diet and Plastic Ingestion in the Blackmouth Catshark *Galeus melastomus*, Rafinesque 1810, in Italian Waters

**DOI:** 10.3390/ani13061039

**Published:** 2023-03-13

**Authors:** Giorgia Zicarelli, Chiara Romano, Samira Gallo, Carmen Valentino, Victor Pepe Bellomo, Francesco Luigi Leonetti, Gianni Giglio, Alessandra Neri, Letizia Marsili, Concetta Milazzo, Caterina Faggio, Cecilia Mancusi, Emilio Sperone

**Affiliations:** 1Department of Chemical, Biological, Pharmaceutical and Environmental Sciences, University of Messina, 98166 Messina, Italy; 2Department of Biology, Ecology and Earth Sciences, University of Calabria, Via P. Bucci, 87036 Rende, Italy; 3Consorzio per il Centro Interuniversitario di Biologia Marina ed Ecologia Applicata “G. Bacci” (CIBM), Viale N. Sauro 4, 57128 Livorno, Italy; 4Department of Environment, Earth and Physical Sciences, Siena University, Via Mattioli 4, 53100 Siena, Italy; 5Environmental Protection Agency of the Tuscany Region (ARPAT), Via Marradi 114, 57126 Livorno, Italy

**Keywords:** *Galeus melastomus*, Pentanchidae, feeding, plastics pollution, Mediterranean Sea

## Abstract

**Simple Summary:**

The blackmouth catshark has a habitat range that spans from the Norwegian seas to Senegal and throughout the Mediterranean Sea, and it is one of the most common sharks in Italian waters. The aim of this work is to investigate, through the analyses of the stomach contents of five populations from the Tyrrhenian and Ionian Seas, the diet of blackmouth catsharks. The analyses showed that the most frequent items were Osteichthyes of the family Myctophidae, except for one population, in which the most common items were Cephalopods and Crustacean of the Decapods order. Plastic debris was also found in all populations analysed and classified by colour and shape. This study ought to increase the knowledge of the feeding ecology of the blackmouth catshark, thus improving the meagre literature about Tyrrhenian and Ionian waters.

**Abstract:**

*Galeus melastomus* is the most common Pentanchidae in the Mediterranean Sea. A scavenger and opportunistic feeder, and despite the wide distribution, little is known about its feeding habits in Italian waters. The main purpose of this study was to investigate the diet of the blackmouth catshark by analysing the stomach contents. The specimens analysed were obtained from five populations of the Tyrrhenian and of the Ionian Seas, collected from a depth between 40 and 700 m. A total of 259 stomachs were analysed. The stomach contents were grouped into macro-categories and identified to the lowest taxonomic level possible. Crustaceans such as *Parapenaeus longirostris*, the Cephalopods *Heteroteuthis dispar* and *Onychoteuthis banksii*, and Osteichthyes, mostly Myctophidae, were identified. Plastic debris was also found among the stomach contents and classified according to its colour and shape. Osteichthyes represent the most abundant item (44%), above all the Myctophidae family, except for the catshark population from Tuscany, in which the most frequent species were Cephalopods, such as *Abralia veranyi* and *Heteroteuthis dispar*. Differences in the plastic debris contents were also observed between the Tuscany population and other populations. These could be explained as a probable consequence of the different depths at which the blackmouth catshark populations were sampled.

## 1. Introduction

Within an ecosystem, each species occupies a certain role called “an ecological niche”, requiring different conditions for survival [1]. For the animal species, the word “ecological niche” also indicates their food requirements by organising biological communities according to trophic levels. These levels represent the ways in which species obtain energy from the environment [2]. The trophic interactions in the marine environment are very complex, so it becomes more appropriate to speak of a trophic web and not a chain, since the same organism may be involved in several trophic levels [3]. Within such interactions, sharks play the role of top predators, placing themselves at the apex of the trophic web [4]. Indeed, sharks control the prey populations of the lower trophic levels, preventing negative impacts on the trophic web. Therefore, knowledge of the trophic ecology of sharks becomes a key point for understanding aquatic communities and their health status. 

The blackmouth catshark *Galeus melastomus* Rafinesque, 1810 is a Pentanchidae inhabiting demersal habitats. It is distributed in the Eastern Atlantic, from Norway to Senegal, and up to the Azores; in the Mediterranean basin, *G. melastomus* is the most abundant species among catsharks [5,6,7] and is commonly found in all Italian seas [8], with the exception of the Adriatic Sea, where it is rare [9]. *G. melastomus* occurs over a wide bathymetric range (55–1400 m), although it is usually found between 300 and 800 m deep along the outer continental shelf and slope habitats, preferring moving backdrops [7,10,11,12,13,14]. Juveniles (young of the year and sub-adult) are generally distributed between 200 and 500 m deep, whereas the adults are concentrated at a depth between 500 and 800 m [15,16]. *Galeus melastomus* is a small oviparous and iteroparous species [17], with reproduction peaks during late spring and summer [15,18]. The embryos feed only on their yolk [19], and the number of eggs is proportional to the female size. As an opportunistic and scavenger shark with a diversified diet [7], *G. melastomus* can reach a maximum registered length of 90 cm [5]. Thanks to its developed eyes with the retina rich in photoreceptors, *G. melastomus* can detect the bioluminescence of its prey [20]. The diet of the blackmouth catshark includes many crustaceans, cephalopods, and fish. Studies on the feeding habits of this species were carried out in the Atlantic Ocean (Cantabrian Sea [13], Portugal [21]); in the Western Mediterranean Sea (Iberian Peninsula [22,23], Gulf of Lions [24], and Western Algerian coastline [25,26]); and Eastern Mediterranean Sea (Aegean Sea [27,28]). These studies showed that blackmouth catsharks feed mainly on crustacean decapods such as *Sergestidae*, euphausiids (*Meganycthipanes norvegica*), cephalopods (e.g., *Todarodes sagittatus* and *Heteroteuthis dispar)*, and the fish belonging to Myctophidae and Gonastomatidae [23]. The main threat to this species is represented by the bycatch, particularly in the traditional deep trawl fishery for blue and red shrimp *Aristeus antennatus* [29,30,31] and giant red shrimp *Aristaeomorpha foliacea*. Despite being a species with low commercial value, occasionally, relatively large individuals are destinated for commercialisation [32]. The blackmouth catshark is considered a Least Concern by the IUCN. Predators of *G. melastomus* are larger sharks [33], and remains of this species have been found in the stomach contents of *Dalatias licha* [34]. 

Nonetheless, one aspect of concern for the survival of the species is the presence of plastics in the environment, which can be ingested.

Since the first documented evidence in 1970 of plastics in the marine environment [35], the discovery of plastics and microplastics in the stomach contents of marine species has become more frequent, and in some cases, it was considered as the main cause of death [36,37,38]. There are several ways by which these plastics are ingested: indirect ingestion, biomagnification, and through the gills, where plastics debris remains trapped [37,39,40]. The Italian waters are affected by plastic pollution [41]. Plastics and microplastics have been found in the stomach contents of several elasmobranch [42]—for example, four demersal species (*Scyliorhinus canicula*, *Squalus acanthias*, *Mustelus asterias*, and *Scyliorhinus stellaris*) in the United Kingdom. Among sharks, analyses on blackmouth catsharks were conducted in the Balearic Islands, highlighting the presence of plastics [43]. In the last decade, a growing body of evidence has reported the presence of plastics in the stomach contents of *G. melastomus* in the Mediterranean Sea [44,45,46,47,48].

Despite the presence of *G. melastomus* in the Mediterranean Sea—in particular, in Italian waters—information about its diet in the area is scarce, fragmentary, or non-existent. The present work aims to provide more insights into the feeding of the blackmouth catshark in a wide area of its range by analysing populations along the Italian Peninsula from the north (Liguria) to the southeast (Gulf of Taranto). 

Taking into account such a wide range of geographic distribution and different bathymetries is essential to provide the most realistic picture of the animal’s adaptation to different ecological conditions. Indeed, the seas of the Italian Peninsula are characterised by a high heterogeneity of environments influenced by the currents, which affect the distribution of the communities.

Moreover, emphasis is placed on the presence of microplastics and plastics with a dimension greater than 5 mm in the stomach contents of *G. melastomus*, indirectly providing important information on the state of pollution of the area studied. 

## 2. Materials and Methods

### 2.1. Collection of Data 

The collection of the samples took place in four areas of the Tyrrhenian Sea (Liguria, Tuscany, Latium, and Calabria) and in one area of the Ionian Sea (Gulf of Taranto) between 38° N and 44° N and between 17° E and 8° E (Figure 1; Table 1). The Tyrrhenian Sea is commonly divided into the Northern Tyrrhenian Sea and Southern-Central Tyrrhenian Sea. The northern part is more heterogeneous from the ecological and morphological points of view. The continental shelf extends up to 150 m. The maximum depth reached is between 2000 and 2200 m, the principal currents originate from the wind, and they are subjected to significant seasonal variations [49]. The geophysical, morphological, and dynamic structures of the Southern-Central Tyrrhenian Sea are one of the more complex of all the Italian peninsulas. There are two principal abyssal plains with a maximum depth between 2900 and 3600 m [50], and there are two important submarine volcanos. In Calabria, the continental shelf is less developed. The Strait of Messina marks the boundary between the Tyrrhenian and the Ionian Seas, where strong vertical gradients, horizontal gradients, and tidal currents originate [51].

The Ionian Sea is the deepest sea of the entire Mediterranean Basin with a mean depth of ca. 2000 m [52]. The Taranto Valley divides the Ionian Sea into two slopes: the western and the south-eastern slopes. The first has a vast continental shelf, whereas the second presents many submarine canyons. The Ionian Sea receives superficial Atlantic waters and Levantine intermediate waters up to 800–900 m depths; deeper waters arrive from the Adriatic Sea [51]. 

The sampling activities took place between October 2020 and August 2021. Samples from Liguria, Tuscany, and Latium were obtained by MEDITS (Mediterranean International Trawl Survey) scientific surveys using a bottom trawl [53,54,55]. Sampling was carried out between 100 and 700 m deep, and the nets remained in the water for 30/60 min, with a total of 100/150 bridle for each haul. The mesh size and the vessel speed were in accordance with the information reported in the MEDITS Handbook [56]. The mesh size of the cod end was 10 mm per side, which corresponds to an approximately 20 mm mesh opening. To ensure the best trawl geometry, the recommended fishing speed was 3 knots on the ground.

Samples from Tyrrhenian Calabria and from Ionian Seas were obtained by local fishermen as a result of the bycatch and was acquired at fish markets at the landings. The samples from Calabria were collected in the Gulf of Santa Eufemia at an average depth of 40 m with a bottom longline from a boat, whereas the Ionian Sea samples were collected in the Gulf of Taranto by means of a small fishing boat with pole and lines at a depth of about 300 m. 

In this work, 302 samples of blackmouth catshark were collected, distributed as follows: 10 from the Ionian Sea, 35 from Calabria, 184 from Latium, 28 from Tuscany, and 45 from Liguria. In the laboratory, the following biometric measurements were taken for each sample: Total Length, TL (±0.1 cm); Fork Length, FL (±0.1 cm); and Weight, W (±1.0 g). The specimens of blackmouth catshark were grouped into three age classes according to their lengths: young of the year (x ≤ 23 cm), sub-adult (23 < x < 34 cm), and adult (x ≥ 34 cm), in agreement with information on the biology of the species [17,18]. The sex was identified by the presence/absence of a clasper, and the maturity stage was determined according to the directive of the MEDITS project [53]. 

The stomach and the spiral valve were removed from the animals and opened under a stereomicroscope to observe the stomach contents [57]. The stomach contents were grouped into four macro-categories: crustacean, mollusc, fish, and other (including plastic remains). For each sample, both the total weight of the stomach contents and the weight for each macro-category were recorded. A total of 259 stomachs were analysed. 

The taxa were classified at the lowest categories possible, depending on the type of analyses performed and the digestion level, with the help of taxonomic guides and scientific articles [58,59,60,61,62,63]. The identification of cephalopods was made by observing the beaks under a stereomicroscope, while fish identification was performed using otoliths. When the species could not be identified, an estimate of the number of cephalopods and fishes was made by using the remains of the lens of the eyes [58,59,60,61,62,63]. 

Plastic remains were documented from all the stomach contents. According to Eriksen et al. [64] and Valente et al. [48], the shape (fibres, fragments, film, or sphere) and the colour (black, blue, green, grey, orange, red, white, or yellow) were noted. All plastic remains were also measured using graph paper and divided into micro-, meso-, and macroplastics following the categories of the Marine Strategy Framework Directive [45,65,66,67,68,69,70], in which the upper boundary of the microplastics was 5 mm and the upper size of the mesoplastics was 25 mm. Plastics < 1 mm were considered microplastics, and macroplastics did not have an upper boundary (>25 mm).

One of the main challenges faced when analysing plastic fibres derives from specimen contamination by exogenous agents, such as airborne fibres. To avoid false negatives, controlled measures have been put in place, in accordance with other studies such as Pedà et al. [45]. Each operator was wearing a lab cotton coat, Petri dishes were covered with carefully washed caps, windows were closed, ventilation systems shut off, and the steel instrumentation was thoroughly washed with distilled water before each dissection.

Following these arrangements, all plastic fibres were considered as true positives (originating from fishing nets) and not from laboratory environmental contamination (e.g., experimenter clothes).

### 2.2. Data Analyses 

The comparison of the diets between the five populations was developed by studying the Frequency Occurrence (FO%; the percentage of stomachs containing at least one item of prey) compared to the total number of stomachs containing prey and the percentage number (N%; the correlation between the total number of prey items) within the totality of stomachs and the total number of prey items inside the stomachs. The values were analysed with the Kruskal–Wallis test. Linear regression was applied to assess a possible correlation between the number of items and the lengths of the animals, as well as plastics ingestion.

To assess the diversity and richness in the considered populations, cluster analyses was used according to Preti et al. [70], Ganesh and Geetha [71], and Janžekovič and Klenovšek [72]. In the cluster analyses, they were performed at the family level in order to normalise the data among the populations, and the population from the Gulf of Taranto was excluded. 

The Simpson diversity index combines the number of species and the relative abundance of the prey type. It measures the probability that two individuals, randomly chosen in a sample, can belong to a different species. The values range from zero to one. 

All the data were analysed using the free software PAST (Paleontological statistics) version 4.0. 

Similar analyses were performed for plastic remains, using colour and shape as the variables. All the plastics for every population were analysed with Kruskal–Wallis and Mann–Whitney tests by correlating colour and shape; the prevalence of each in males and females was also recorded.

## 3. Results

The mean total lengths and weights of the sharks for each region were 44.5 ± 6.1 cm and 252.1 ± 104.9 g for the Tyrrhenian Calabria samples, 50.9 ± 3.6 cm and 353.3 + 99.9 g for the Gulf of Taranto, 25.5 ± 8.3 cm and 44.0 ± 61.6 g for Latium, 45.1 ± 1.6 cm and 215.5 ± 28.4 g for Tuscany, and 31.1 ± 8.6 cm and 86.2 ± 77.9 g for Liguria. Females had a mean total length of 32.6 ± 12.4 cm and mean weight of 131.4 ± 139.8, while males had a mean total length 31.3 ± 10.5 cm and mean weight of 95.6 ± 84.4. Six stomachs were empty, whereas the others contained at least one prey item. A total of 48 taxa were identified (Table 2). The diet in all the studied populations of *G. melastomus* was diversified. The analyses of N% and FO% highlighted a higher consumption of the crustacean *Parapenaeus longirostris*; the cephalopods *Abralia veranyi*, *Onychoteuthis banksii*, and *Heteroteuthis dispar*; and fishes belonging to the family Myctophidae. 

The linear regression analysis revealed an extremely significant correlation between the number of stomach items and body length (r^2^ = 0.06 and *p* < 0.0001).

The Kruskal–Wallis test was used to compare the food classes in samples from each region. The results showed that, in the Gulf of Taranto and Tuscany, *G. melastomus* preys mainly upon molluscs (KW = 12.8882, *p* = 0.003 and KW = 24.75, *p* < 0.0001, respectively), whereas, in the other three populations (Tyrrhenian Calabria, Latium, and Liguria), it mainly consumes fishes (***) (KW = 46.58, *p* < 0.0001; KW = 231.39, *p* < 0.0001; and KW = 46.02, *p* < 0.0001). The same test was used to investigate food preference between males and females of blackmouth catsharks, with the former showing a preference for fish, followed by cephalopods, and then crustaceans (KW = 151.33 and *p* < 0.0001) and greater amounts of ingested plastic. By applying the Kruskal–Wallis test on the Latium population grouped into age classes, it has been observed that adults eat fish and cephalopods in the same quantity (KW = 90.5 and *p* < 0.0001), whereas sub-adults and young of the year eat mainly fish (KW = 71.5, *p* < 0.0001 and KW = 175.07, *p* < 0.0001, respectively). No significant differences in the number of ingested items (KW = 5.4, *p* = 0.06), nor in the number of ingested plastics (KW = 3.4, *p* = 0.1), were found relative to the age classes (Figure 2). 

In all analysed populations, plastics were found in the stomach contents, with the Tyrrhenian Calabria population exhibiting the highest FO%. 

The mean number of plastics items observed in the stomachs was 2.0 for Ionian Calabria, 1.8 for Tyrrhenian Calabria, 0.8 for Latium, 2.1 for Liguria, and 0.6 for Tuscany; the results of the Kruskal–Wallis were KW = 16.714 and *p* = 0.002. The Kruskal–Wallis test on the shape and colour for each region revealed the following: the Tyrrhenian Calabria population showed a high percentage of transparent, black, and grey plastics (KW = 34.420 and *p* < 0.0001) and of fragments and fibres (KW = 18.232 and *p* = 0.0004); the Gulf of Taranto population exhibited a higher percentage of white plastics (KW = 0.231 and *p* = 0.4) and fragments (KW = 4.583 and *p* = 0.2); the Latium population contained transparent and white plastics (KW = 38.190 and *p* < 0.0001) and fragments (KW = 38.162 and *p* < 0.0001); and the Liguria population showed a high percentage of black plastics (KW= 25.776 and *p* = 0.001) and fragments (KW = 9.290 and *p* = 0.02), while the Tuscany population had black and blue plastics in higher percentages (KW = 12.655 and *p* = 0.1) and fibres (KW = 7.781 and *p* = 0.05). All results are summarised in Figure 3 (colours) and Figure 4 (shapes).

The analysis between the populations and the sizes of the plastics highlighted that the Tyrrhenian Calabria population had the highest mean size of plastics (5.5 ± 11.8 mm, KW = 13.753, and *p* = 0.008). The other regions had an average size of 4.3 ± 4.7 mm for the Gulf of Taranto, 2.9 mm for Latium (±8.78 mm) and Liguria (±37.1 mm), and 2.0 ± 10.8 mm for Tuscany. The size structures of all plastic remains found in the five populations are summarised in Table 3.

Females contained a higher mean number (1.7) of plastics than males (0.6), but the Mann–Whitney test did not report any significant difference (U = 7640.5; U’ = 9099.5; *p* = 0.2). Similar results were obtained, with the dimensions of plastics being higher in females (4.1 ± 25.9 mm) than in males (2.4 ± 9.3 mm) (U = 7553. 0; U’ = 9187.0; *p* = 0.1). Both males and females showed the same dominant colour order: first black (35.48% for males and 33.94% for females), then white (21.50% for males and 25.69% for females) and transparent (17.20% for males and 22.48% for females) plastics above all others (KW = 41.813 and *p* < 0.000; KW = 34.578 and *p* < 0.0001). The females showed a high percentage of fragment-shaped plastics (KW = 37.814 and *p* < 0.0001) and males a high percentage of fragments and fibres (KW = 31.610 and *p* < 0.0001). Males and females ingested approximately an equal number of items.

No correlation was found by the linear regression analysis between the number of plastics ingested and the lengths of the animals (r^2^ = 0.008, *p* = 0.1). However, a significant correlation was detected between plastic size and the length of the animal. The size of the plastics was proportional to the body length. Larger plastics were found in larger animals, with a linear regression between r^2^ = 0.15 and *p* = 0.01.

To assess the diversity and richness in each population we applied, the cluster analyses and diversity indices (Simpson diversity, Shannon, and Evenness) and data were normalised using family taxa for all populations.

A cluster analysis identified the presence of the same prey species in the five populations (Figure 5). Comparing the three macro-categories in four populations, excluding the population from the Gulf of Taranto, the dendrograms displayed higher similarities between the Tuscany and Liguria populations (0.75) and, then, with the Latium one. The population from Calabria was the more distant, based on diet composition. However, there was not a great similarity, ranging from 0.75 to 1.000.

As expected, Latium sub-adults and adults exhibited a similar diet, with differences only in young of the year (Figure 6), a very common observation in elasmobranchs that changes their diet with age.

Regarding plastics ingestion, the picture is completely different. The similarities of plastics by colour and shape in the four populations show the formation of two clusters: one with Calabria and Latium and the other one with Tuscany and Liguria (Figure 7).

## 4. Discussion

Albeit, several papers in the literature have described *Galeus melastomus* feeding behaviour, they present either incomplete data or data limited to a specific geographical area in the Mediterranean Sea. For these reasons, other areas, such as those considered in this paper, still need to be studied in more detail. 

The most commonly found family in the stomach contents was the Myctophidae, in accordance with the published literature [13,23,73], confirming that blackmouth catsharks may use visual predation, taking advantage of prey bioluminescence to better detect food. 

To analyse the prey item macro-categories (Crustacea, Mollusca, and Osteichthyes), they were combined by family level taxa and based on the entire diet data pool; the results indicate major differences in diets among the five populations.

The cluster Tuscany–Liguria is linked at a value of approximately 0.75. One of the reasons for the similarity between these two populations could be their geographical proximity; indeed, the other population closest to these two was the Latium one. The differences highlighted for the Calabrian populations could be due to pregnant females in Calabria swimming up into shallow water for reproduction and laying egg cases on the bottom, as previously suggested [16,31], despite normally living at greater depths. 

Linear regression confirms and highlights a known correlation [21,25]: age and body size both affect dietary change, and a higher percentage of items in animals correlates with a longer body length. All three macro-categories are present in the age classes, but as the animal grows, the species become more diversified. Species of small dimensions such as *Abralia veranyi* and *Heteroteuthis dispar* were common prey in YOY and sub-adults, in accordance with a previous work [21]. On the contrary, animals of bigger dimensions such as *Todarodes sagittatus* were rare in the diet of YOY and frequently or exclusively found in the adults’ diet.

In more detail, the data analyses support the hypothesis that blackmouth catshark’s diet modifications are seasonal [27,74,75] and vary with both age and bathymetric variations. The data indicate low similarity between populations regarding predation on cephalopods. In the N%, the Tuscany and Gulf of Taranto populations are paired, and in these populations, the samples were all adults and sampled at higher depths; the Latium population had the lowest value. A similar pattern was obtained for crustacean and fish consumption. Contrary to the published data showing crustaceans as the favourite prey of juveniles—in particular, *Parapenaeus longirostris* and other decapods [13,26]—in the Tyrrhenian populations, crustacean decapods were preyed upon mostly by adults living at great depths compared to the individuals that lived at shallower depths (Calabria) or to the juveniles (Latium).

Differences in the diets between males and females could be linked to an uneven sampling regime and not to a real pattern. Even the distribution within the age classes was uneven and could influence the diet dynamics.

Tests such as Kruskal–Wallis and Mann–Whitney confirmed the results about the preference of *G. melastomus* for fish of the Myctophidae family. Molluscs (*H. dispar*, *A. veranyi,* and *O. banksii*) and fishes (*Electrona risso* and *Scomberesox saurus*) were statistically significant prey items. The results were similar to those found for the Gulf of Lions [20] but not in other works, such as Fanelli et al. [23] or Darna et al. [25], in which the main items were crustacean such as euphausiids. Comparing the results obtained with the one for the Cantabrian Sea [13], there is a clear difference between the preferred prey of blackmouth catsharks: in the Tyrrhenian and Ionian Seas, the fish found in the stomach contents were small (Myctophidae such as *Electrona risso* (6.72% FO), with a maximum length of 8 cm), whereas, in the Cantabrian Sea, the prey were larger, such as *Merluccius merluccius* (140 cm) or *Trachurus trachurus* (40 cm). In the Cantabrian Sea, the most common prey were euphausiids, mysids, and benthopelagic shrimps, whereas cephalopods were preyed upon by lesser spotted dogfish. Dietary changes could be linked to changes in prey availability and not only to seasonal variations, as observed by Anastasopoulou et al. [27]; during autumn, there was a higher species diversity (carbide shrimps, fish, cephalopods, and prawns) compared to the summer, when most predators preyed on fish and cephalopods, despite the fish declining in the past years. 

Very often, studies about the diet of *G. melastomus* are correlated with other deep sea Chondrichthyes families, such as Scyliorhinidae and Chimaeridae, or species such as *Etmopterus spinax* [13,23,24,26,28] living in the same area. From this perspective, between velvet belly lantern sharks and blackmouth catsharks, there is a partial diet overlap, with both species having mesopelagic habits and a preference for Myctophidae [23]. Species commonly found in the stomach contents of blackmouth catsharks such as *Heteroteuthis dispar* (39.92% FO) were barely present in the stomach contents of velvet belly lantern sharks [23]. On the contrary, the diets of the lesser spotted dogfish and the blackmouth catsharks were observed to be more similar despite *Scyliorhinus canicula* being a benthic feeder and *Galeus melastomus* a demersal feeder. Both species feed on particular crustaceans (such as *Parapenaeus longirostris* [26]) and fish. While decapods such as *Pasiphaea sivado* and *Pasiphaea multidentata* are preferred preys of blackmouth catsharks, *S. canicula*, on the other hand, feeds primarily on shrimps such as *Alpheus glaber* and *Solenocera membranacea* near to the seabed [13]. Crustacean species such as *Dalatias licha*, although covering the same geographic area as *G. melastomus*, have an upper trophic position with the predation of others sharks [34,76]. Two other Mediterranean chondrichthyans, *Raja asterias* and *Raja clavate*, occupy a similar trophic niche as the blackmouth catshark *clavate*, and they both feed on fish and crustaceans—in particular, the superfamily Penaeidae [77], which was also found in the stomach contents of *G. melastomus.*

The steady variations of the *G. melastomus* diet are evident by the results obtained in this work and in the literature. Nonetheless, the ecological impact of the blackmouth catshark on benthic and demersal communities is not known.

Lastly, the ingestion of plastics did not represent any pattern or intentional predation by *Galeus melastomus;* the consumption of plastics occurs randomly [43] and with differences linked mostly to bathymetry. Blue or black plastics of the fibre type, considered as internal to the stomach before it was opened, were also noted by Browne et al. [78] and Napper & Thompson, [79], probably derived from cloths and synthetic clothes that have been through washing machine cycles and reached the seabed through wastewater treatment plants. For that reason, sharks captured at the greatest depths (Tuscany) present fibres in the highest percentages. The same results were obtained by Alomar & Deudero [43], with 86.36% of fibres found in the stomachs of *G. melastomus* sampled at 600 m. However, white, transparent, and black plastic types that represent the main colours found in other papers [45,46] originate from vials and plastic bags, and they can be found both at shallow depths and in the water column. The ingestion of plastics by Myctophidae fish species that have a nictemeral migration was observed in the literature [80]; therefore, the presence of plastics in blackmouth catshark stomach contents could be due to the ingestion of Myctophidae with plastics in their stomachs.

Elasmobranchs feeding on large fish such as *Dalatias licha* [34] also brings a possible transfer of plastics up the trophic chain. These might indirectly be due to human intervention, because, despite having a low commercial interest [32], *G. melastomus* falls under food fraud. The skin is removed to make it unrecognisable, and they are exchanged for lesser spotted dogfish (*Scyliorhinus canicula*) and for spurdog (*Squalus* spp.) and sold. This practice represents a danger to humans, because microplastics could be the carrier of biological agents [81,82].

## 5. Conclusions

In conclusion, *Galeus melastomus* have a diet that constitutes similar macro-categories of prey phyla in all the Italian waters, but the percentages change in relation to age classes, bathymetry, geographical area, and season variations. We could define the blackmouth catshark as an accidental plastic feeder, which may be ingested directly or indirectly. This study comparing five populations provides an overview of the diet of one of the most common sharks in the Ionian and the Tyrrhenian Seas. It improves the meagre literature about the Tyrrhenian Sea and the Ionian Sea, but more studies will be needed to completely understand the food ecology and the feeding behaviour of these small deep sea sharks.

## Figures and Tables

**Figure 1 animals-13-01039-f001:**
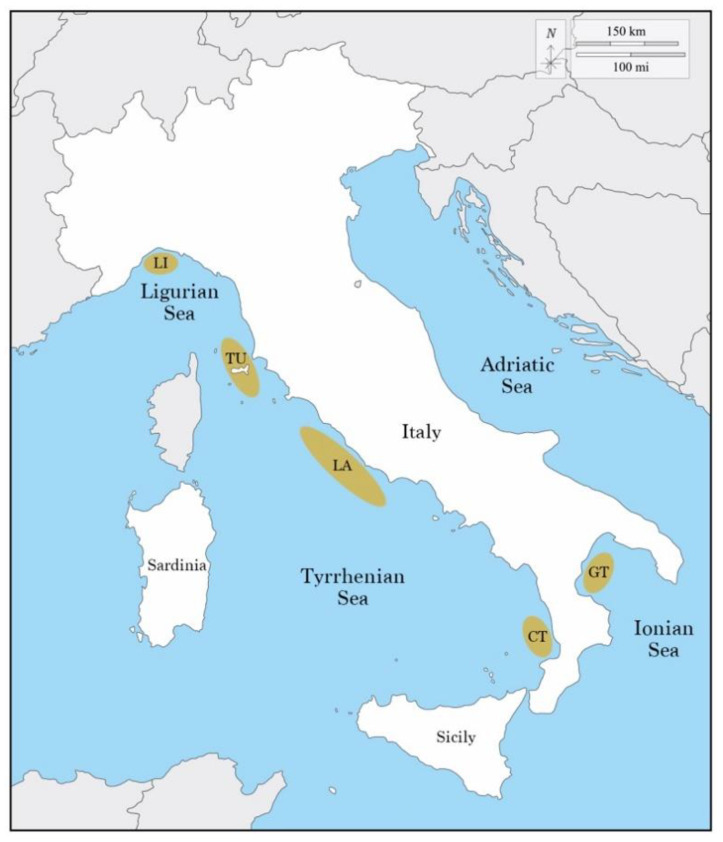
Locations of the sampling areas (LI: Liguria; TU: Tuscany; LA: Latium; CT: Calabria; GT: Gulf of Taranto). Details of the sampling sites are reported in Table 1.

**Figure 2 animals-13-01039-f002:**
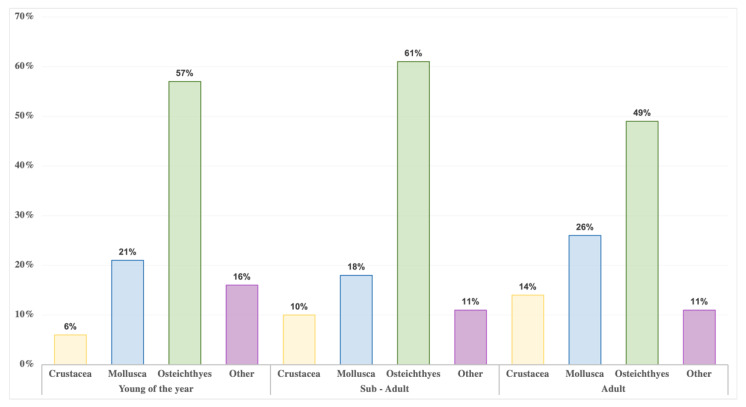
Percentage of items (N%) found in the stomach contents per age classes in the Latium population.

**Figure 3 animals-13-01039-f003:**
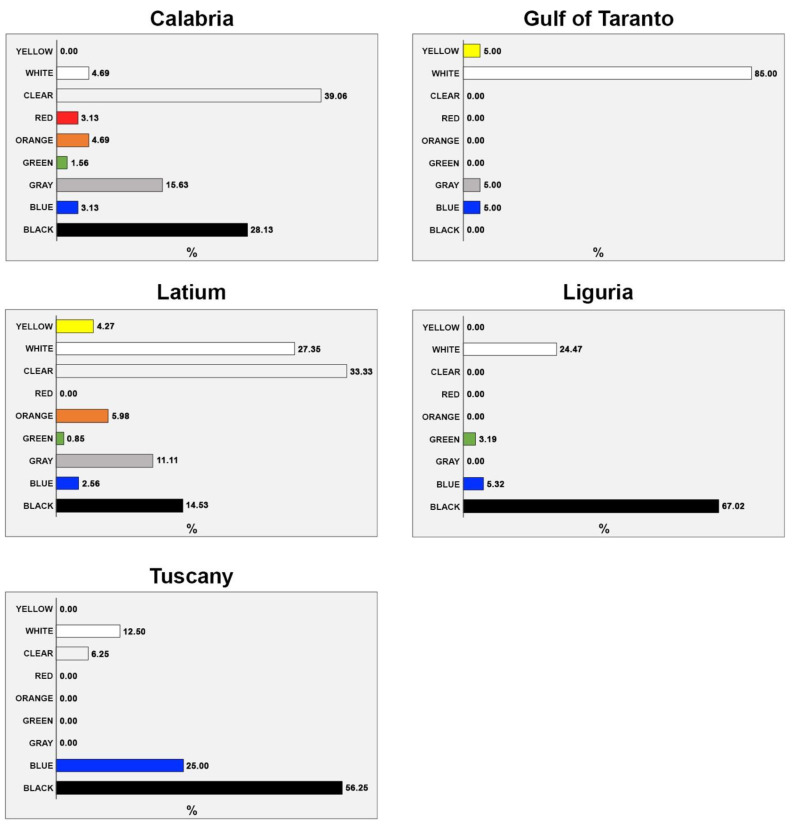
Frequency distribution by colour of plastic fragments found in the stomachs of blackmouth catsharks in the five populations studied. The colour of the bars corresponds to that of the plastics.

**Figure 4 animals-13-01039-f004:**
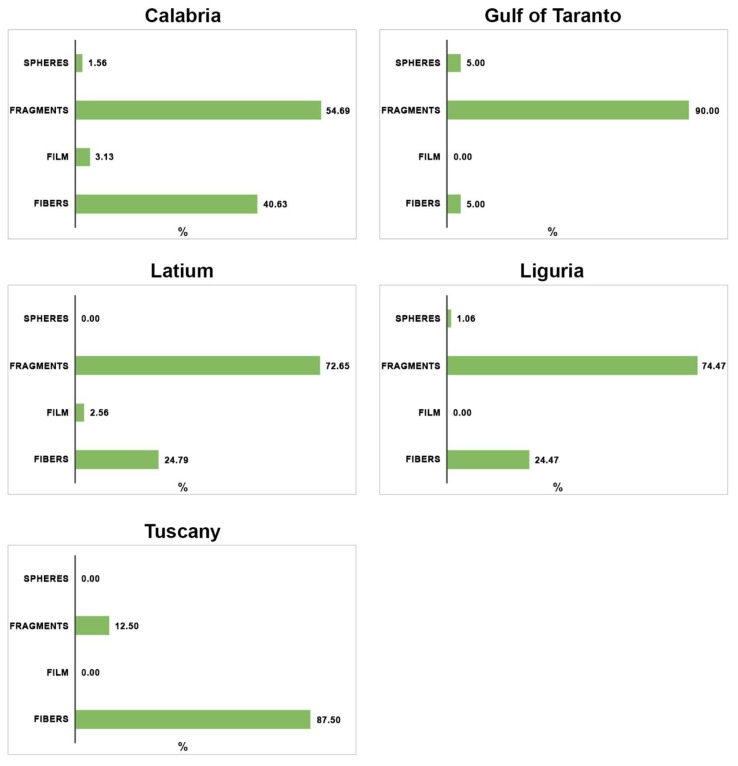
Frequency distribution by shapes of plastic fragments found in the stomachs of blackmouth catsharks in the five populations studied.

**Figure 5 animals-13-01039-f005:**
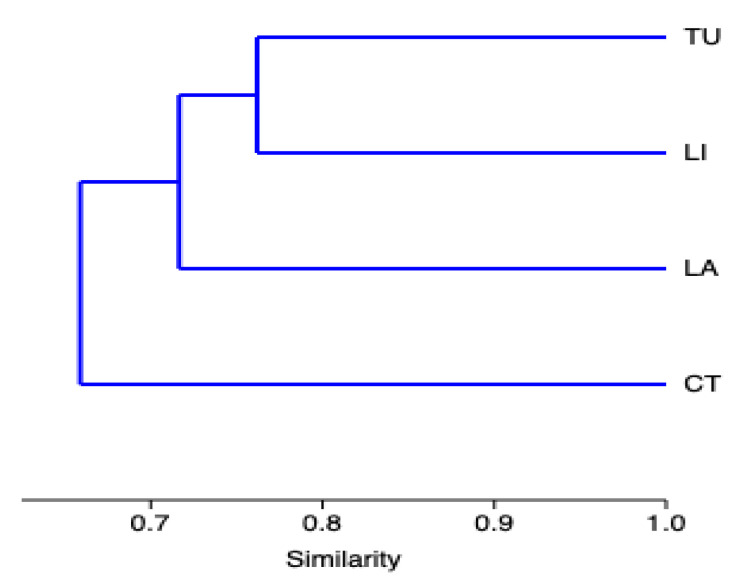
Cluster analyses. The greatest similarity occurs for populations whose values are close to zero. The comparison is between the three macro-categories analysed at the family level with high similarity between the populations of Tuscany (TU) and Liguria (LI). The acronyms LA and CT refer to Latium and Calabria.

**Figure 6 animals-13-01039-f006:**
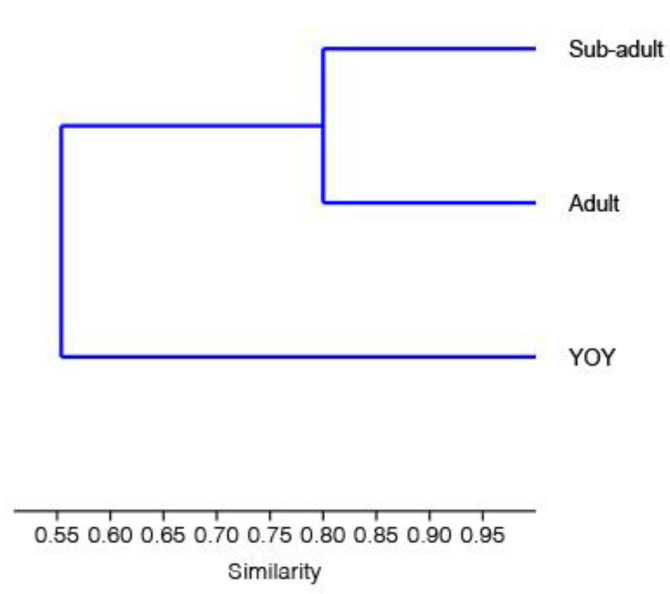
Cluster analyses in the population of Latium between age classes with greater similarities in the diets of sub-adults and adults.

**Figure 7 animals-13-01039-f007:**
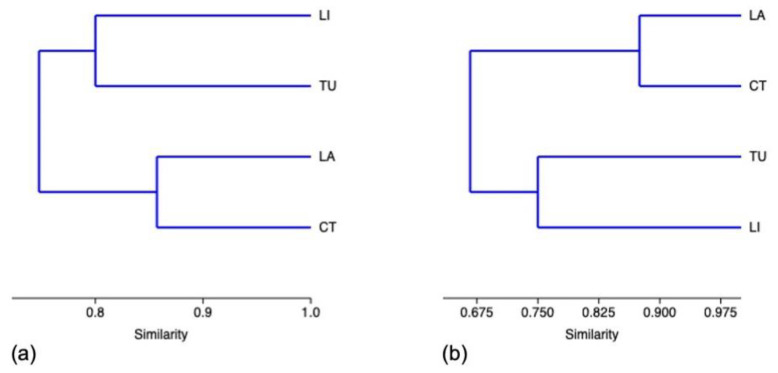
Cluster analyses for plastic debris. (**a**) Similarities between plastics by shape and (**b**) the similarities between plastics by colour.

**Table 1 animals-13-01039-t001:** Summary of the regional location, longitude, latitude, date, number of individuals and depth for sampling sites.

Region	Sampling Sites Coordinates		Date	Number of Individuals	Depth
	Longitude	Latitude			(m)
Liguria	43°48′450″	9°38′270″	20 October 2020	16	412.5
	44°06′680″	9°32′050″	19 October 2020	6	392
	44°15′740″	8°40′840″	18 October 2020	4	582
	43°58′520″	8°17′270″	17 October 2020	19	471.5
Tuscany	43°35′310″	9°35′520″	20 October 2020	26	634.5
	42°15′190″	10°11′220″	14 October 2020	2	251
Latium	41°22′780″	12°17′580″	3 November 2020	6	282
	41°29′740″	12°08′500″	1 November 2020	29	497.5
	41°27′580″	12°10′980″	1 November 2020	34	432.5
	41°48′110″	11°48′990″	31 October 2020	12	476.5
	42°04′770″	11°09′790″	30 October 2020	103	347
Calabria	38°94′1037″	16°12′5332″	7 November 2020	6	40
	38°94′1037″	16°12′5332″	11 February 2021	12	40
	38°94′1037″	16°12′5332″	12 April 2021	17	40
Gulf of Taranto	40°04′1865″	16°88′4223″	2 February 2021	2	300
	40°04′1865″	16°88′4223″	12 August 2021	8	300

**Table 2 animals-13-01039-t002:** Summary of all the species found in the stomach contents for each sampling area and the total N% and FO%. The number indicates how many items of that species were found in the stomach contents. The number of N% and FO% are in %.

Taxa	Calabria	Tuscany	Latium	Liguria	Gulf of Taranto	N%	FO%
Crustacea							
DECAPODA							
*Carcinus aestuarii*					1	0.06	0.40
Natantia							
Superfamily: Penaeidae			8	1		0.54	2.37
*Aristaeopsis edwardsiana*		1				0.06	0.40
*Aristeus antennatus*			6			0.36	1.19
*Parapenaeus longirostris*	1		7	4	1	0.78	4.35
Family: Pasiphaeidae							
*Pasiphaea* sp.				1		0.06	0.40
*Pasiphaea sivado*			6	1		0.42	2.77
*Pasiphaea multidentata*		1		1		0.12	0.79
Family: Sergestidae							
*Eusergestes arcticus*		1				0.06	0.40
ISOPODA	1	1	4		1	0.42	2.77
Unidentified crustacean	10	17	53	40	2	7.28	39.53
Mollusca							
Family: Brachioteuthidae							
*Brachioteuthis riisei*					3	0.18	0.79
Family: Chiroteuthidae							
*Chiroteuthis veranii*		2	2			0.24	1.58
Family: Enoploteuthidae							
*Abralia veranyi*	14	1	9	9	1	2.04	11.58
Family: Loliginidae							
*Loligo forbesii*			1		2	0.18	0.79
Family: Histioteuthidae							
*Histioteuthis* spp.	1	1			2	0.24	1.19
*Histioteuthis bonnellii*	1	3		1	1	0.36	1.58
*Histioteuthis reversa*	5	11	2	5		1.38	5.93
Family: Ommastrephidae							
*Illex coindetii*		1	1	2	2	0.36	2.37
*Todarodes sagittatus*	1	16	1	1	9	1.68	6.32
Family: Onychoteuthidae							
*Ancistroteuthis* spp.		1			4	0.30	0.79
*Ancistroteuthis lichtensteinii*	2	4	1			0.42	2.77
*Onychoteuthis* sp.					1	0.06	0.40
*Onychoteuthis banksii*	2	17	3	3	5	1.80	8.70
Family: Sepiolidae			4			0.24	0.79
*Heteroteuthis dispar*	77	22	81	23	18	13.29	39.92
*Rossia macrosoma*	1	1				0.12	0.79
*Sepietta* sp.					1	0.06	0.40
Octopoda							
*Argonauta argo*			1			0.06	0.40
*Eledone cirrhosa*				1		0.06	0.40
Unidentified cephalopods	29		97	35	9	11.13	45.06
Osteichthyes							
Family: Caristiidae	1					0.06	0.40
Family: Gadidae			1			0.06	0.40
Family: Nettastomatidae							
*Nettastoma melanorum*			1			0.06	0.40
Ordine: Myctophiformes	5					0.30	0.79
Family: Myctophidae	3		4			0.42	1.58
*Benthosema glaciale*	1		2			0.18	1.19
*Ceratoscopelus* sp.			1			0.06	0.40
*Ceratoscopelus maderensis*	2		3	1	4	0.60	2.77
*Diaphus* sp.			5	1		0.36	2.37
*Diaphus rafinesquii*	1	1			1	0.18	1.19
*Electrona risso*	19		6			1.50	6.72
*Lampanyctus* sp.				1		0.06	0.40
*Lampanyctus crocodilus*	3					0.18	1.19
*Myctophum punctatum*			2	1		0.18	1.19
Family: Scomberesocidae							
*Scomberesox saurus*	9		1			0.60	3.16
Family: Stomiidae			4	1		0.30	1.98
*Stomias boa*		1	1	1		0.18	1.19
Unidentified fishes	142	30	517	121	24	50.15	87.75
Others							
Annelida	1	1	1			0.18	1.19
*Turdus merula*			1			0.06	0.40
Echinodermata				1		0.06	0.40
*Sylvia curruca*			1			0.06	0.40

**Table 3 animals-13-01039-t003:** Percentage of the size structure of plastics divided into three size categories: microplastics, mesoplastics, and macroplastics.

Plastics Percentage			
Microplastics		Mesoplastics	Macroplastics
x < 1 mm	1 < x < 5 mm	5 < x < 25 mm	25 mm > x
2.6%	32.2%	64.8%	0.3%

## Data Availability

Data are available on request due to restrictions, e.g., privacy or ethical.

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
