# Peer review of "Diet and Plastic Ingestion in the Blackmouth Catshark Galeus melastomus, Rafinesque 1810, in Italian Waters"

_animals, 2023, doi:10.3390/ani13061039_

Round 1
Reviewer 1 Report (New Reviewer)
I have evaluated the originality, significance and technical quality of the study, and the length and clarity of the review paper, which reference number and title are given below.
Reference Number: animals-2197814
Title: “Feeding and plastic ingestion in the Blackmouth catshark Galeus melastomus (Rafinesque, 1810) from the central Mediterranean Sea”
The study is well prepared, fluently written in English language and results are well presented and discussed with sufficient references.
The present work concentrates on feeding and plastic ingestion in the Blackmouth catshark in the Mediterranean region. The study provides important information regarding the stomach content of Blackmouth catshark that is found to accidentally consume plastic by direct or indirect ingestion. Findings provide important information for the understanding of the food ecology and the feeding behavior of these sharks.
In brief, I suggest to accept this article in its present form without any further improvement.
For your information
Best regards,
The Reviewer
Author Response
We thank the reviewer for appreciating our paper
Reviewer 2 Report (New Reviewer)
The manuscript describes stomach content in 259 individuals of the blackmouth catshark among 302 that were captured.
I am surprised to find no authorization number or ethics committee opinion to authorize this study in which over 300 individuals are killed.
There is a lack of clear information regarding the geography of the study sites. For example, it would have been interesting to note where the Tyrrhenian Sea and the Ionian Sea are located and to show the separation between the northern and southern Tyrrhenian Sea. The catching sites are not clearly identified since in Table 1 there are 16 catching sites, in Figure 1 there are 8 and at the end in Table 2 there are only 5. There should be more homogeneity in the locations, and it should be clearer.
The catches were made between October and August and it would have been interesting to establish a date effect on the results obtained.
The number of tests that are presented on page 8 is very important, are they all really useful? By doing all these tests that do not really provide information, there is a risk of having a large number of false positives or false negatives linked simply to the increase in the number of tests.
The tests that should be done are tests for which there is a hypothesis that is really tested and not a set of tests simply because we have the data.
Concerning the statistical analyses, there is an ambiguity in the text on how the tests are done; for example, X2 tests are indicated as being done on percentages. I would like to be sure that it is not percentages that were tested but numbers. From the text it is not clear.
The authors state on line 424 that the presence of plastic does not represent an intentional pattern or predation but that the consumption of plastic is more reflective of what is present in the environment, especially at the bathymetry. Couldn't this be the equivalent for the predation data and could it be related to a catch period effect? It would be interesting at least to discuss this point.
We can note a bias in the beginning of the introduction linked to the geographical position of the teams that carried out this work since in line 46 it is indicated that the Mediterranean Sea is a sea that has been studied a lot because of its historical importance. The Mediterranean Sea is indeed important from a historical point of view for the Mediterranean region, but not on a global scale! This first sentence is too Mediterranean-centric.
Some minor changes to be made:
Line 21: add a space between 1400 and m
Line 87: remove "Also"
Line 122: Show "Tarento Valley" on the map
Line 145: Put a non-breaking space between 1.0 and g
Table 2: Is this the number of individuals for whom the number of items was found or the total number of items?
Line 206: Change "ad" to "and
Table 3: Why is the synthesis only shown in the Latium population?
Line 370: remove the , in ,:
Author Response
The manuscript describes stomach content in 259 individuals of the blackmouth catshark among 302 that were captured. I am surprised to find no authorization number or ethics committee opinion to authorize this study in which over 300 individuals are killed.
The chondrichthyan specimens enrolled in the present work were obtained from commercial fisheries and from the MEDITS programs (trawl fishing for the scientific survey). The activity was conducted with the observation of the Regulation of the European Parliament and the Council for fishing in the General Fisheries Commission for the Mediterranean (GFCM) Agreement area and amending Council Regulation (EC) No. 1967/2006. Moreover, an ethic statement was sent to the Academic Editor.
There is a lack of clear information regarding the geography of the study sites. For example, it would have been interesting to note where the Tyrrhenian Sea and the Ionian Sea are located and to show the separation between the northern and southern Tyrrhenian Sea. The catching sites are not clearly identified since in Table 1 there are 16 catching sites, in Figure 1 there are 8 and at the end in Table 2 there are only 5. There should be more homogeneity in the locations, and it should be clearer.
In table 1 were indicated all the sampling sites (16) with their coordinate, some of these are very close one each other ex. Houl 65 and 109 and for that reason, fewer points were shown on the map (Fig.1). Regardless, all the sampling was made in five places of the Italian Peninsula (Liguria, Tuscany, Latium, Tyrrhenian Calabria and Gulf of Taranto). The analyses were not conducted on the individual haul. The map was updated for showing the division between the Tyrrhenian Sea and the Ionian Sea and highlighting the Italian Region as suggested by the Academic Editor, and table 1 was modified changing the information and maintaining just the information about the Italian Region.
The catches were made between October and August and it would have been interesting to establish a date effect on the results obtained.
Establishing a seasonal effect on the data it’s difficult because the sampling was made in more seasons just for the Calabria region (Autumn, Winter and Spring), but no substantial change in the diet composition was seen during the identification.
The number of tests that are presented on page 8 is very important, are they all really useful? By doing all these tests that do not really provide information, there is a risk of having a large number of false positives or false negatives linked simply to the increase in the number of tests.
The tests that should be done are tests for which there is a hypothesis that is really tested and not a set of tests simply because we have the data.
Concerning the statistical analyses, there is an ambiguity in the text on how the tests are done; for example, X2 tests are indicated as being done on percentages. I would like to be sure that it is not percentages that were tested but numbers. From the text it is not clear.
We have reduced, simplified and rearranged all the statistics in the paper. The X2 analyses and the diversity index were deleted from the analyses conducted on the samples, and the cluster analysis was modified by omitting the Gulf of Taranto population, since we had few individuals from that population.
The authors state on line 424 that the presence of plastic does not represent an intentional pattern or predation but that the consumption of plastic is more reflective of what is present in the environment, especially at the bathymetry. Couldn't this be the equivalent for the predation data and could it be related to a catch period effect? It would be interesting at least to discuss this point.
Certainly, as written in the discussion, bathymetry is one of the reasons why the diet of G. melastomus varies, and as indicated in literature even the seasonality influenced it; but, for this paper establishing a seasonality in diet changing is difficult. Having seasonality in only one region, we preferred to maintain the focus on the diversity of the diet of the blackmouth catshark when we consider different geographic areas in the same sea (Tyrrhenian Sea) and at different bathymetry excluding, just for now, the seasonality.
We can note a bias in the beginning of the introduction linked to the geographical position of the teams that carried out this work since in line 46 it is indicated that the Mediterranean Sea is a sea that has been studied a lot because of its historical importance. The Mediterranean Sea is indeed important from a historical point of view for the Mediterranean region, but not on a global scale! This first sentence is too Mediterranean-centric.
The introduction about the Mediterranean Sea, as suggested by the Academic Editor was deleted and replaced with an introduction about the importance of the trophic ecology of the sharks and then focusing the attention on G. melastomus. This paragraph, besides, was added under the request of another reviewer during the previews revision of the manuscript.
Some minor changes to be made:
Line 21: add a space between 1400 and m
The correction was made to the text.
Line 87: remove "Also"
The word was deleted as requested.
Line 122: Show "Tarento Valley" on the map
As specified above the map was modified and information such as the division of the seas, the region and the Taranto Valley was added.
Line 145: Put a non-breaking space between 1.0 and g
The correction was made to the text.
Table 2: Is this the number of individuals for whom the number of items was found or the total number of items?
In Table 2 were written the items found inside the stomach content for that species. Since it was not clear, the specification was added in the description of the table.
Line 206: Change "ad" to "and
The correction was made to the text
Table 3: Why is the synthesis only shown in the Latium population?
Because the analysis of the age classes was conducted just for the Latium population that has a high number of samples. Table 3, besides, was modified in a graph in accordance with the suggestion by the Academic Editor.
Line 370: remove the , in ,:
The correction was made to the text.
Round 2
Reviewer 2 Report (New Reviewer)
The authors have adequately answered to the previous comments.
Author Response
Many thanks to the reviewer for his help to improve the MS
This manuscript is a resubmission of an earlier submission. The following is a list of the peer review reports and author responses from that submission.
Round 1
Reviewer 1 Report
Review for the paper "Feeding habits of the Blackmouth catshark Galeus melastomus (Rafinesque, 1810) from the central Mediterranean Sea, with emphasis on plastic ingestion" by Giorgia Zicarelli, Chiara Romano, Samira Gallo, Carmen Valentino, Victor Pepe Belomo, Francesco Luigi Leonetti, Gianni Giglio, Alessandra Neri, Letizia Marsili, Concetta Milazzo, Caterina Faggio, Cecilia Mancusi, Emilio Sperone submitted to "Animals".
General comment.
The authors conducted several catches in different periods of the year and at different sites of the Mediterranean Sea in order to study the diet of the Blackmouth catshark Galeus melastomus proposing to improve our knowledge on the feeding of this less-studied fish. The idea is good and could contribute to this topic, since there are few reports dealing with the stomach content of the species. They revealed the composition of the main prey items in five locations and concluded that Mychtiophidae were preferably ingested by four populations and one population fed mainly on mollusks and crustaceans. They also revealed the presence of microplastics in the stomachs of the fish. Standard methods to catch samples and to process the data were used in the study. Statistical methods seem to be adequate and correctly used. The main results are illustrated with relevant Figures and Tables. In the current form, the ms is of local interest and must be improved considerably to make it interesting for an international audience. Some parts of the ms need to be expanded and clarified, especially Introduction, Results and Discussion.
Major criticisms of the ms:
(1) Introduction.
I feel that a baseline description of the study area would be appropriate in the beginning of the Introduction.
The authors must provide more data regarding biology, ecology, range and distribution of the species. Pay more attention to the life cycle, population and size structure of the species, trophic position in the marine food webs.
I think that a short description of the previous results regarding feeding of the Blackmouth catshark should be included in this section.
The authors must highlight clearly the novelty of their study.
(2) Material and methods.
Sampling procedures must be described more carefully especially trawling (mesh size, vessel speed during trawling.
Figures 4-6 indicate the results of cluster analyses. However, I found no description of this method in the Material and methods. Please, update the M&M section.
(3) Results.
The authors must give data regarding weights of Galeus melastomus individuals collected in different locations.
The authors must give a table showing the contribution of the prey taxa in the stomach contents of each size class of Galeus melastomus.
Figures 4-6 indicate results of cluster analyses. However, I found no any description of this method in the Material and methods. Please, update the M&M section.
Please, provide data, if you have, regarding the size structure of the microplastic particles.
(4) Discussion.
The authors must compare their results with other sites and habitats.
Discuss your results from the ecological point of view, namely the possible ecological impact of the species on the benthic and fish communities, ecological role of the species, and possible influence on fishery due to preying on commercial fishes.
Pay more attention to the role of the species in the food web in transferring the microplastics.
Specific remarks.
Reconsider the title as follows: Feeding of the Blackmouth catshark Galeus melastomus (Rafinesque, 1810) in the central Mediterranean Sea, with emphasis on plastic ingestion.
Also, I suggest replacing ‘feeding habits’ with ‘feeding’ throughout the entire ms.
L23. Consider replacing "from Tyrrhenian" with "from the Tyrrhenian".
L24. Consider replacing "From analyses emerged" with "The analyses showed".
L26, 264. Consider replacing "Crustacean" with "Crustacea".
L39. Provide contribution (percentage, range and mean) of the main prey items to the diet of Galeus melastomus.
L159. Consider replacing " diversified " with " diverse".
L196-197. Consider replacing "From these analyses it emerges that for the Tyrrhenian Calabria the amount of transparent" with "In the Tyrrhenian Calabria, the amount of transparent".
L205. Consider replacing "In Latium" with "In Latium,".
L222. Consider replacing "in the 5 populations " with "in the five populations ".
L262. Consider replacing "similarit" with " similarity".
L268. Consider replacing " Latium " with " Latium individuals".
L305. Consider replacing " levelled to family " with " were combined by family level".
Author Response
Response to Reviewer 1 Comments
Review for the paper “Feeding habits of the Blackmouth catshark Galeus melastomus (Rafinesque, 1810) from the central Mediterranean Sea, with emphasis on plastic ingestion” by Giorgia Zicarelli, Chiara Romano, Samira Gallo, Carmen Valentino, Victor Pepe Belomo, Francesco Luigi Leonetti, Gianni Giglio, Alessandra Neri, Letizia Marsili, Concetta Milazzo, Caterina Faggio, Cecila Mancusi, Emilio Sperone submitted to “Animals”.
General comment.
The authors conducted several catches in different periods of the year and at different sites of the Mediterranean Sea in order to study the diet of the Blackmouth catshark Galeus melastomus proposing to improve our knowledge on the feeding of this less-studied fish. The idea is good and could contribute to this topic, since there are few reports dealing with the stomach content of the species. They reveled the composition of the main prey items in five locations and concluded that Mychtiopidae were preferably ingested by four populations and one population fed mainly on mullusks and crustaceans. They also revealed the presence of microplastics in the stomachs of the fish. Standard methods to catch samples and to process the data were used in the study. Statistical methods seem to be adequate and correctly used. The main results are illustrated with relevant Figures and Tables. In the current form, the ms is of local interest and must be improved considerably to make in interesting for an international audience. Some parts of the ms need to be expanded and clarified, especially Introduction, Results and Discussion.
Major criticism of the ms:
- Introduction
I feel that a baseline description of the study area would be appropriate in the beginning of the Introduction.
L46. According to the suggestion of the reviewer, a description of the study area was added at the beginning of the Introduction but more detailed information was given in materials and methods during the description of the collection of data.
The authors must provide more data regarding biology, ecology, range and distribution of the species. Pay more attention to the life cycle, population, and size structure of the species, trophic position in the marine food webs.
L57 to L72. Data about the biology, ecology, range and distribution was added in the introduction accordingly, and new and more information was given about the life cycle, population, size structure and trophic position of the Galeus melastomus.
I think that a short description of the previous results regarding feeding of the Blackmouth catshark should be included in this section.
L80. Following the suggestion of the reviewer information about the previous results of the feeding habits was provided.
The authors must highlight clearly the novelty of their study.
L103. The aims of the study were rewritten and the novelty was highlighted clearly according to the suggestion of the reviewer.
- Material and methods
Sampling procedures must be described more carefully especially trawling (mesh size, vessel speed during trawling).
L143. The required information was added to the text, especially information about the MEDITS programme (trawling samples) from which most samples came.
Figures 4-6 indicate the results of cluster analyses. however, I found no description of this method in the Material and methods. Please, update the M&M section.
L192. The description required was added.
- Results
The authors must give data regarding weight of Galeus melastomus individuals collected in different locations.
L220. The mean weight was calculated for every population of Galeus melastomus taken into consideration and added to the text accordingly.
The authors must give a table showing the contribution of the prey taxa in the stomach contents of each size class of Galeus melastomus.
L258. A table (Table 3) with the percentage of occurrence of prey taxa (Crustacea, Mollusca, Osteychties and Others) was given in the test. The table refers to the Latium population because is the one in which the division in age classes is more substantial.
Figures 4-6 indicate the results of cluster analyses. However, I found no description of this method in the Material and methods. Please, update the M&M section.
L192. As required by the reviewer, the information about the cluster analyses was added to the material and methods.
Please, provide data, if you have, regarding the size structure of the microplastic particles.
L179 and L323. The size structure of the plastics, as required, was added to the text and summarized in the table 4. According to Eriksen et al., (2014) plastics was divided in three different group: micro, meso and macroplastics.
- Discussion
The authors must compare their results with other sites and habitats.
L439. The comparison between other sites of the Mediterranean Sea and other habitats (Cantabrian Sea) in which the Galeus melastomus is present was provided in the discussion accordingly the request.
Discuss your results from the ecological point of view, namely the possible ecological impact of the species on the benthic and fish communities, ecological role of the species, and possible influence on fishery due to preying on commercial fish.
L456. A little discussion from an ecological point of view was given in the text. It is difficult quantifying the influence of the species on commercial fish due to the lack of information about it in the literature.
Pay more attention to the role of the species in the food web in transferring the microplastics
L475. Information about the transferring the microplastics in the food web was added to the text.
Specific remarks.
Reconsider the title as follows: Feeding of the Blackmouth catshark Galeus melastomus (Rafinesque, 1810) in the central Mediterranean Sea, with emphasis on plastic ingestion.
L2. As suggested the title was changed omitting the word “habits”.
Also, I suggest replacing “feeding habits” with “feeding” throughout the entire ms.
Following the suggestion “feeding habits” was replaced with “feeding” throughout the entire ms.
L23. Consider replacing “form Tyrrhenian” with “from the Tyrrhenian”.
L23. The change was done in the text.
L24. Consider replacing “From analyses emerged” with “The analyses showed”.
L24. The phrase “From analyses emerged” was replaced accordingly.
L26,264. Consider replacing “Crustacean” with “Crustacea”.
L26,343. The change was made accordingly.
L39. Provide contribution (percentage, range and mean) of the main prey items to the diet of Galeus melastomus.
L39. As suggested the percentage of the main items was added, unfortunately, the range and the mean couldn’t be calculated because the size of the otoliths was not taken and the prey was all digested.
L196-197. Consider replacing “From these analyses it emerges that for the Tyrrhenian Calabria the amount of transparent” with “In the Tyrrhenian Calabria, the amount of transparent.”
L268. Following the suggestion, the change was made in the text.
L205. Consider replacing “In Latium” with “In Latium,”
L277. The comma was added accordingly.
L222. Consider replacing “In the five populations” with “in the five populations”.
L294. The number “5” was replaced with the word “five” accordingly.
L262. Consider replacing “similarit” with “similarity”.
L340. The error was corrected in the text accordingly.
L268. Consider replacing “Latium” with “Latium individuals”.
L346. The word “individuals” was added at the ms accordingly.
L305. Consider replacing “levelled to family” with “were combined by family level”.
L395. In according with the suggestion the change was made in the text.
Reviewer 2 Report
Dear authors,
unfortunately you manuscript "Feeding habits of the Blackmouth catshark Galeus melastomus (Rafinesque, 1810) from the central Mediterranean Sea, with emphasis on plastic ingestion" is not considerable for publication in the special issue of the journal "ANIMALS".
I would like to give you some advices to try to modify your manuscript before resubmitting it to a lower IF journal.
Despite the considerable sampling effort, there are several criticisms in this work.
First of all, a deep revision of English language must be done. Several syntax mistakes are present in the manuscript.
Introduction is too scant and does not support well the investigation you conducted. In the results and in the title, you put emphasis also on plastic ingestion, but you never mentioned plastics in the intro.
MeM Section is poor supported by references, and especially the part on stomach contents is not well justified (you use a self-citation from a conference proceeding and this reference is not focused on stomach contents).
Since Galeus melastomus is a species that lives in the deep, is it an error that in Calabrian sites (Table 1) specimens were collected at 40 m of depth??
Since you considered three size classes, did you have enough specimens per size class to determine stomach contents? considering this, you should have at least 15 specimens per size class per site to obtain relevant results...
Data analysis is insufficient.
Avoid the use of words like "CALA" that is not an English word (or explain its meaning)
This is a special issue on trophic webs. Comparisons with feeding habits of other species are not present in the work, so in my opinion this is another reason to reject the paper.
Improve quality and completeness of figures (graphs without axis titles and so on).
The title is "feeding habits", the main focus is on stomach contents but the paper lacks of figures on these results.
You analyzed only stomach contents, but this analysis conducted alone in the way you did, is not sufficient to define feeding habits of this species or to enhance the importance of your work on this well-studied species.
Table 3: too many numbers after comma. Moreover, in international journals, the use of comma as decimal separator is not allowed.
I suggest a complete revision the entire work (maybe additional experiments) and re-writing of the manuscript.
Author Response
Response to Reviewer 2 Comments
Dear authors,
Unfortunately you manuscript “Feeding habits of the Blackmouth catshark Galeus melastomus (Rafinesque, 1810) form the central Mediterranean Sea, with emphasis on plastic ingestion” is not considerable for publication in the special issue of the journal “ANIMALS”.
I would like to give you some advices to try to modify your manuscript before resubmitting it to a lower IF journal.
Despite the considerable sampling effort, there are several criticisms in this work.
First of all, a deep revision of the English language must be done. Several syntax mistakes are present in the manuscript.
We regret that the work seemed to the reviewer inappropriate for publication in ANIMALS. However, we have taken into consideration the comments of all the reviewers which, in our opinion, have really improved the work. We hope that now our contribution can be better considered. Tongue revision was performed by a native English-speaking marine biologist.
Introduction is too scant and does not support well in the investigation you conducted. In the results and in the title, you put emphasis also on plastic ingestion, but you never mentioned plastics in the intro.
L90. In accordance with the suggestion of reviewer 1 in the introduction, more information about biology, distribution, life cycle and results of previous work on Galeus melastomus was added to the text. Following the suggestion was added a part in which we talk about plastics and plastics remains in stomach content.
MeM Section is poor supported by references, and especially the part on stomach contents is not well justified (you use a self-citation from a conference proceeding and this reference is not focused on stomach contents).
In the cited works the processing methodologies used in our laboratories are described. We have replaced the citation from a conference proceeding with that of a published paper in which stomach contents have been analysed.
Since Galeus melastomus is a species that lives in the deep, is it an error that in Calabrian sites (Table 1) specimens were collected at 40 m of depth??
L60. No, is not an error, in the Gulf of Santa Eufemia (Calabrian Tyrrhenian sites), there is a population of Galeus melastomus that lives at 40 depth. Furthermore, as specified in the introduction of the ms, in literature (Ebert et al., 2020) the distribution range of G. melastomus goes from 55 to 1400m.
Since you considered three size classes, did you have enough specimens per size class to determine stomach contents? Considering this, you should have at least 15 specimens per size class per site to obtain relevant results…
L258. We consider the size class in particular for the Latium population where the samples were more than 100 and well distributed in class size with over 15 specimens for size.
Data analysis is insufficient.
It seems strange to us that the data analysis is insufficient. In addition to having calculated Frequency Occurrence and N%, we have applied statistical tests such as chi-square and Kruskal-Wallis to the contents and plastics. Furthermore, we calculated three diversity indices (Simpson Diversity Index, Shannon, and Evenness) and performed cluster analysis to evaluate similarities in the composition of the diet of the investigated populations and in the composition of ingested microplastics. We therefore believe that our analyses cannot in any way be considered insufficient
Avoid the use of words like “CALA” that is not an English word (or explain its meaning).
L130,151. Both in Figure 1 and in table the word “CALA” was changed to the word “Houl” following the suggestion.
This is a special issue on trophic webs. Comparisons with feeding habits of other species are not present in the work, so in my opinion this is another reasons to reject the paper.
L382. Following the comment in the discussion was added and made a comparison between the feeding habits of the G. melastomus and other species that live in the same habitat.
Improve quality and completeness of figures (graph without axis titles and so on).
All figures are complete
The title is “feeding habits”, the main focus is on stomach contents but the paper lacks of figures on these results.
L2. The title was changed deleting the word “habits” following the suggestion of the first reviewers focusing the attention more on the feeding of Galeus melastomus.
You analysed only stomach contents, but this analysis conducted alone in the way you did, is not sufficient to define feeding habits of this species or to enhance the importance of your work on this well-studied species.
We changed the title and focus of the work from feeding habit to simply feeding
Table 3: too many numbers after comma. Moreover, in international journals, the use of comma as decimal separator is not allowed.
L375. In accordance with the comment, the numbers after the comma were reduced to two and the comma was substituted with the full stop.
I suggest a complete revision the entire work (maybe additional experiments) and re-writing of the manuscript.
We reviewed the work according to all reviewers' comments.
Round 2
Reviewer 1 Report
The authors have revised the paper according to my comments.
Reviewer 2 Report
Dear authors,
I am really sorry to say that I still consider your manuscript not publishable on this journal (and, in the present form, in any scientific journal). There is a substantial problem in the number of samples collected from each area (not comparable data). In the Introduction, you state that the Strait of Messina divides the Mediterranean Sea into West and East Med, and this completely not true.
Galeus melastomus lives also in the South Adriatic, in the Bari Canyon.
The added parts in the introduction and all along the paper still need a deep English revision (sentences are not always comprehensible)
In the Intro you talked about 253 samples but in the results about 259..
In methodologies, you cannot use self-citations such as that of Reinero et al. because Reinero et al. took the used method from another paper, so Reinero et al. did not show a new methos for preservation of parasites. We all understand the need of making a research group grow, but this is not a correct method to do it.
You present mean values of TL etc but were are the standard deviations?
"By correlating the N% and FO% with the chi-squared test": X2 does not correlate variables. This is not the correct test to analyse these data. A multivariate analysis had to be run.
You compare the diets of animals with a substantial different TL and this should not be done.
Carcinus estuari is a really shallow item. how could it be found in stomach contents?
Aristaeopsis edwardsiana is a stricly atlantic species which presence is not signaled in Mediterranean Sea.
The most interesting part is that on the presence of plastic in stomach contents (even if for examples the sentences from line 446 to line 453 do not have any sense and the discussion should be completely re-written as the entire manuscript).
I suggest the authors to focus on the presence of plastic in the stomachs of a deep species sampled along a wide longitudinal gradient and to re-submit the paper elsewhere (it is not considerable for this special issue).